

# A novel clustered-based detection method for shilling attack in private environments

Ihsan Gunes

Department of Computer Technologies, Eskisehir Technical University, Eskişehir, Turkey

## ABSTRACT

The topic of privacy-preserving collaborative filtering is gaining more and more attention. Nevertheless, privacy-preserving collaborative filtering techniques are vulnerable to shilling or profile injection assaults. Hence, it is crucial to identify counterfeit profiles in order to achieve total success. Various techniques have been devised to identify and prevent intrusion patterns from infiltrating the system. Nevertheless, these strategies are specifically designed for collaborative filtering algorithms that do not prioritize privacy. There is a scarcity of research on identifying shilling attacks in recommender systems that prioritize privacy. This work presents a novel technique for identifying shilling assaults in privacy-preserving collaborative filtering systems. We employ an ant colony clustering detection method to effectively identify and eliminate fake profiles that are created by six widely recognized shilling attacks on compromised data. The objective of the study is to categorize the fraudulent profiles into a specific cluster and separate this cluster from the system. Empirical experiments are conducted with actual data. The empirical findings demonstrate that the strategy derived from the study effectively eliminates fraudulent profiles in privacy-preserving collaborative filtering.

## INTRODUCTION

As a result of the rapid growth of the technologies underlying the Internet, electronic commerce has grown increasingly prevalent. Over the past few years, online shopping has become increasingly popular. Customers are assisted in making appropriate product selections from a collection of available products by means of recommender systems, which have been created in recent years. Collaborative filtering, also known as CF, is quickly becoming one of the most popular types of recommender systems (*Bobadilla et al., 2013*). It is possible that customers do not want their product preferences and the things they score to be made public. Methods of privacy-preserving collaborative filtering, also known as privacy-preserving collaborative filtering (PPCF), have been developed (*Polat & Du, 2003*) in order to safeguard these kinds of individually held preferences. When it comes to PPCF schemes, the objective is to deliver suggestions with a level of accuracy that is acceptable while protecting users' privacy.

The term "randomized perturbation" refers to a technique that is frequently used in PPCF algorithms as a means of protecting users' privacy. The data are masked using this method, which involves the addition of noise data. Therefore, data collectors who store

Corresponding author
Ihsan Gunes, igunes@eskisehir.edu.tr

disguised data are unable to learn the actual rates, but they continue to make accurate forecasts. In order to generate random numbers, a Gaussian or uniform distribution that has a mean of zero and a standard deviation ($\sigma$) is typically utilized. In order to conceal the rated and/or unrated products, the cells corresponding to some of the uniformly randomly picked unrated items have been filled with random integers.

Users with malicious intentions who are attempting to manipulate the results of CF and PPCF systems are able to launch attacks against these systems by inserting fake profiles into their databases. In most cases, the goal of these attacks is to either raise the popularity of the product that is being targeted (a "push attack") or decrease that popularity (a "nuke attack"). Shilling attacks are a sort of cyberattack that have been documented by *Burke et al. (2005)*, *Lam & Riedl (2004)*, and *O'Mahony et al. (2004)*. The research conducted by *O'Mahony et al. (2004)* and *Burke et al. (2005)* demonstrates that CF systems are susceptible to shilling attacks. According to *Gunes, Bilge & Polat (2013)* and *Bilge, Gunes & Polat (2014)*, not only are CF approaches susceptible to these kinds of attacks, but so are a variety of PPCF systems. That is to say, these kinds of attacks have the potential to have a considerable impact on the accuracy of the estimated predicts made by PPCF systems. As a result, it is of the utmost importance to identify these varieties of attacks and lessen the damage they cause in order for recommendation systems to operate well. *Chirita, Nejdl & Zamfir (2005)*, *Burke et al. (2006)*, *Bhaumik et al. (2006)*, *Mehta (2007)*, *Li & Luo (2011)* and *Zhang & Zhou (2014)* are some of the researchers who created and implemented various detection approaches to CF algorithms for the purpose of determining fake profiles.

However, there has been limited research conducted on the identification of fake profiles in PPCF algorithms. PPCF systems now incorporate masked data to address privacy concerns. Detecting shilling attacks becomes challenging in this scenario. Currently, all transactions are required to be conducted using masked data. As the system masks fake profiles, it may become challenging to identify them. Attack profiles, being generated by a specific algorithm, tend to exhibit a high degree of similarity. Detection methods aim to categorize or cluster attacks based on the similarities in their profiles. Nevertheless, when attack profiles are masked, their similarity to one another diminishes, making it challenging to identify them. In this study, we utilize the ant colony clustering detection method for PPCF schemes. In the existing literature, ant colony clustering algorithms have been employed to address accuracy, sparsity and scalability issues in non-private environments. The utilization of the ant colony clustering algorithm as a detection method represents a significant contribution to the field, providing a novel approach to addressing these challenges.

The reason we chose the ant colony clustering algorithm is because it has not yet been utilized for detecting shilling attacks. Nevertheless, they have the ability to discover optimal or almost optimal solutions for clustering problems. The ACO method employs dispersed agents that emulate the behavior of actual ants in order to discover the most efficient route for optimal clustering solution. These agents utilize pheromone trail data to generate solutions and employ local search operations to enhance those solutions. The algorithm generates a number of cluster distribution solutions equal to the number of

agents and try to continuously improve by selecting the best solutions. The method systematically generates new solutions, executes local search operations, and updates the pheromone trail matrix for a certain number of iterations. The solution's quality is assessed based on the objective function's value, which reflects the optimal or nearly optimal arrangement of pieces into clusters.

In order to accomplish this objective, we initially adapt the ant colony clustering technique to be compatible with PPCF methods, and subsequently carry out experiments utilizing authentic data. The process of attacking PPCF algorithms involves the utilization of six distinct attack models that have been developed in the past (*Gunes, Bilge & Polat, 2013*).

The contributions of this article in general can be summarized as follows:

1) The ant colony clustering-based detection technique is employed to identify shilling profiles on masked data in the databases of PPCF systems.
2) In order to determine the results of the studies, various tests were performed using two real data sets to evaluate the effectiveness of the detection methods on PPCF.
3) This study compares the detection methods employed with other approaches previously utilized in the literature.

## RELATED WORK

In their study, *Chirita, Nejdl & Zamfir (2005)* conducted the initial research on identifying shilling profiles by examining the characteristics of user profiles. They examined the most basic offensive models including random and average strategies. Their approach proves effective when dealing with assault profiles that exhibit a high level of density, but it fails to yield positive results when confronted with profiles that display a significant degree of sparsity. In their study, *Burke et al. (2006)* examined various attributes obtained from user profiles to determine their efficacy in detecting attacks. Their study demonstrates that a machine learning sorting technique including features generated from attack models outperforms a more commonly used detection system. To identify attack patterns, one can employ variable selection using principal component analysis (PCA) (*Mehta, 2007*; *Mehta, Hofmann & Nejdl, 2007*; *Mehta & Nejdl, 2009*). The PCA approach involves computing either the correlation or co-variation values between all users. The data matrix is subsequently arranged in accordance with that computation. The correlation values between the attack profiles exhibit a significant degree of similarity, although the co-variation correlation values are minimal. The technique is limited to dense user-item matrices as PCA is not able to handle null values.

*Zhang & Wang (2020)* proposed a method for detecting group shilling attacks in online recommender systems. The authors utilized the bisecting k-means clustering algorithm to identify attack groups by analyzing rating tracks and calculating suspicious degrees. Experimental results showed that the proposed method outperforms baseline approaches. Their study contributed to the field by addressing the detection of group attacks and providing insights for enhancing the security of recommender systems. *Zhang et al. (2020)*

introduced an innovative approach to identify group shilling attacks in collaborative recommender systems. This method diverges from conventional approaches that concentrate on individual attackers by taking into account the collusive behaviors exhibited by a collective of attackers that collaborate to control the system. The suggested methodology employs graph embedding and clustering methodologies to examine user rating patterns and detect dubious clusters. The suggested strategy outperformed baseline approaches in real-world datasets, as shown by experimental findings.

In their study, *Hao et al. (2023)* introduced an innovative method for identifying hybrid assaults in recommender systems. Conventional detection approaches frequently encounter difficulties in recognizing these attacks, which involve a combination of model-generative shilling attacks and group shilling attacks. The method described use graph convolutional networks (GCN) to extract user data and effectively characterize both types of attacks. A two-stage detection strategy is created, which includes the process of partially tagging user profiles and training a detector based on Graph Convolutional Networks (GCN). The experimental findings have shown the efficacy of the proposed strategy in detecting hybrid attacks. This extensive survey delves into the realm of shilling assaults in recommendation systems, offering valuable insights regarding attack models, detecting features, and detection algorithms (*Sundar et al., 2020*). Additionally, they examined the inherent characteristics of implanted profiles that are manipulated by detection algorithms, providing insights into an uncharted domain. In addition, it provides a concise overview of current advancements in resilient algorithms, assaults on multi-criteria systems, and collaborative filtering techniques that rely on inherent feedback. The authors' objective was to furnish a thorough comprehension of shilling attacks and their detection, presenting significant insights for researchers and practitioners in the domain.

*Cai & Zhang (2021)* proposesd a three-stage detection method for identifying group shilling attacks in recommender systems. By considering strong lockstep behaviors among group members and group behavior features, the proposed approach aims to improve the precision in detecting group shilling attacks. The method involves constructing a weighted user relationship graph, finding dense subgraphs, and using clustering to detect shilling groups. Experimental results on Netflix and Amazon review datasets demonstrate the effectiveness of the approach, with F1-measures reaching over 99% and 76%, respectively. *Wang et al. (2022)* introduced a technique for identifying shilling groups in online recommender systems with a graph convolutional network (GCN). The suggested solution overcomes the constraints of conventional methods by autonomously extracting characteristics for shilling group detection, hence obviating the necessity for manual feature engineering. The approach represents the dataset as a graph, enhances user characteristics by principal component analysis, and employs a three-layer graph convolutional network (GCN) with a neighbor filtering mechanism to classify users. The approach effectively detects shilling groups by studying irregular co-rating relationships between users and identifying target products evaluated by attackers. The experimental findings exhibited exceptional classification accuracy and detection effectiveness across several datasets. The proposed methodology provides a more effective and all-encompassing technique for identifying shilling groups in online recommender systems.

In the existing literature, ant colony clustering algorithms have generally been used to address the accuracy, sparsity and scalability issues in non-specialised environments. *Nadi et al. (2011)* proposed FARS. Web server log data are analysed using FARS to extract user preferences in online platforms. User groups are created *via* ant-based clustering. Ant-based algorithms are essential for optimal results. After the suggestion process, each cluster's pheromone is updated for future use. Accuracy and recall measures measure suggestion precision and completeness. Based on their research, the recommended user grouping technique should increase recommendation accuracy. *Liao, Wu & Wang (2020)* proposed an improvement to the ant colony-based CF algorithm to enhance its performance. This upgrade includes a preliminary phase of user clustering based on pheromone-based user preferences. The number of users exceeds the algorithm's pheromone representation.

Ant colony optimization (ACO) course recommendations have been made from *Sobecki & Tomczak (2010)*. ACO has been shown to solve several optimization challenges. The authors demonstrated ACO's ability to predict university students' final grades. The Trust-based Ant Recommender System (TARS), introduced by *Bedi & Sharma (2012)*, employs dynamic trust among users and ant colony principles to identify the most optimal and compact neighbourhood for effective recommendations. The authors posit that providing additional information about trust graph power and degree of connection, items recommended, and number of neighbours in predicting ratings can enhance active users' decision-making.

In the literature, there are numerous studies on PPCF schemes that can provide suggestions in privacy environments. *Bilge & Polat (2013)* describe the procedure for performing k-means clustering using collaborative filtering (CF) algorithms while ensuring the protection of user privacy. *Luo et al. (2022)* created a clustering-based recommender system that is exceptionally efficient and safeguards user privacy. Homomorphic encryption safeguards user data while CF produces recommendations. The system employs secure clustering to partition data into clusters prior to providing recommendations in order to mitigate information overload. Empirical investigations demonstrate that the suggested approach exhibits efficacy, scalability, and precision in generating suggestions. *Ben Horin & Tassa (2021)* presented distributed mediation-based secure multi-party protocols for privacy-preserving collaborative filtering (PPCF). It gave PPCF a realistic setting and lets users choose between vendors. The study showed that distributed mediation is more secure and faster than single-mediator protocols. The article described the protocols, shows their performance, and discusses related work, emphasizing distributed mediation's advantages over prior art. PRS, developed by *Kashani & Hamidzadeh (2020)*, uses anonymous data conversion and trust-weighted criteria to address security concerns and reduce error rates in collaborative filtering systems. Trust data was converted from perturbation-based chaos to confidential data and clustered using fuzzy c-ordered means and particle swarm optimization. Classification error rates and privacy preservation were better with the proposed method in experiments.

However, few studies in the literature focus on identifying attacks against privacy-preserving recommender systems. *Gunes & Polat (2016)* specifically examined the

identification of shilling attacks in privacy-preserving collaborative filtering systems. Shilling attacks involve the addition of counterfeit profiles to the system with the intention of manipulating the results and thereby diminishing the system's accuracy. The researchers analyzed four attack-detection techniques to identify and eliminate fabricated profiles created by six shilling assaults on manipulated data. Empirical trials are done to assess the efficacy of various detection approaches. The article additionally contrasts the altered detection approaches with the preexisting methods employed in non-private settings. In summary, they provided valuable perspectives on identifying shilling assaults in privacy-preserving collaborative filtering systems. *Yilmazel, Bilge & Kaleli (2019)* introduced a new strategy for detecting shilling attacks in recommender systems that are disseminated in an arbitrary manner. The protocol is based on categorization. The protocol facilitates the identification of malevolent profiles while safeguarding the privacy of the parties involved in collaboration. The research emphasizes the susceptibility of current detection approaches when utilized on dispersed data and suggests a resolution that enables parties to sustain collaboration while identifying shilling attacks. The approach has been shown to produce high detection rates through empirical evaluations utilizing real-world preference data. The authors concluded by addressing potential areas of future research in privacy-preserving distributed collaborative filtering algorithms for material that is distributed in an arbitrary manner. *Bilge, Batmaz & Polat (2016)* proposed a novel method for identifying shilling attacks in PPCF schemes. The approach relies on a bisecting k-means clustering technique, in which attack profiles are gathered within a leaf node of a binary decision tree. *Yilmazel, Bilge & Kaleli (2019)* introduced a new strategy for detecting shilling attacks in recommender systems that are disseminated in an arbitrary manner. The protocol is based on categorization, facilitating the identification of malevolent profiles while safeguarding the privacy of the parties involved in collaboration. The approach has been shown to produce high detection rates through empirical evaluations using real-world preference data.

The scarcity of studies in the literature about shilling attack detection in PPCF systems underscores the significance of our novel approach.

## PREMILINARIES

### Shilling attacks on disguised data

RPT can be utilized to create efficient privacy applications. *Agrawal & Srikant (2000)* propose these techniques. The process of random perturbation (RPT) involves the addition of a randomly generated value (r) to the private data (x) in order to obfuscate it. The generation of random numbers follows a predetermined distribution and assigns values to a database in the form of x + r. Recommendation systems utilize aggregated data. This enables them to provide recommendations based on aggregated perturbation data. PPCF schemes ensure the safeguarding of privacy by preventing servers from acquiring knowledge of the actual ratings and rated items. Perturbed data can be obtained by introducing random values to rates. Users have the ability to input arbitrary values for randomly chosen unrated items. The generation of random values is accomplished by using a Gaussian or uniform distribution with a mean ($\mu$) of zero and a standard deviation

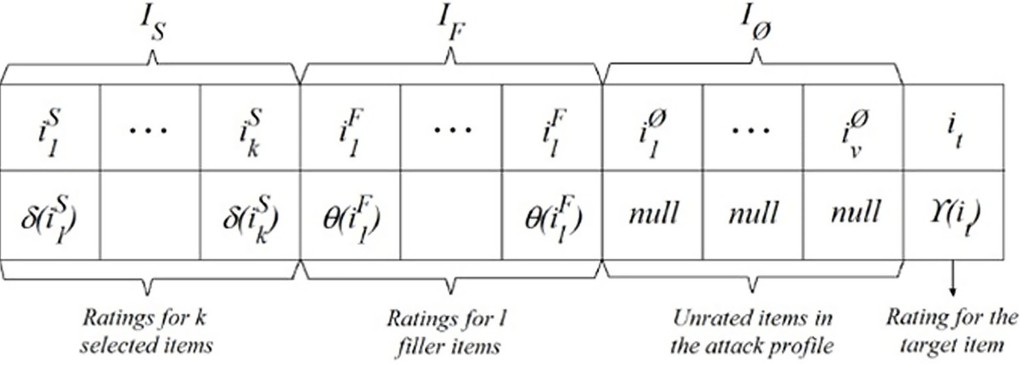

**Figure 1  General form of an attack profile.**

($\sigma$) (*Polat & Du, 2003*). The PPCF scheme begins by normalizing ratings using z-scores. The server establishes the maximum values for $\sigma$ (sigma) and $\beta$ (beta). Define $\beta_{max}$ as the upper limit for the proportion of unrated items that will be assigned random values. It provides users with self-awareness. User $u$ can select values for $\sigma_u$ from $[0, \sigma_{max}]$ and $\beta_u$ in range $[0, \beta_{max}]$. Data disguising, also known as data obfuscation or data anonymization, refers to the process of altering or transforming data in order to protect its confidentiality and privacy.

The act of introducing fraudulent profiles renders the manipulation of any CF system effortless. Attackers may seek to manipulate item forecasts or compromise the recommender system in order to diminish the accuracy of predictions. Figure 1 displays a standard attack profile using a ratings vector of m dimensions, as described by *Mobasher et al. (2007)*. In Fig. 1, the set ls represents a collection of chosen objects that possess specific properties, as decided by the attacker using $\delta$ as a rating function. The filler items, denoted as lf, are chosen at random from the database. The attack profile utilizes a rating function $\theta$ to assign ratings. In order to introduce bias, a specific object is selected and subjected to a rating function $\gamma$. Unrated items are denoted as $I_{\emptyset}$.

Shilling attacks, in general, aim to alter projected predictions. Within PPCF schemes, individuals obfuscate their personal data prior to uploading it onto CF platforms. Hence, executing conventional shilling attacks on PPCF systems becomes challenging. Attackers must adapt their typical shilling tactics to account for the presence of concealed ratings in PPCF schemes. *Gunes, Bilge & Polat (2013)* modified well-known shilling attacks to make them applicable to concealed databases. According to *Gunes, Bilge & Polat (2013)*, attackers must choose between a uniform or Gaussian distribution for random number generation. Furthermore, $\sigma$ values are chosen at random from a uniform distribution within the range $(0, \sigma_{max}]$ for each attack profile before creating counterfeit profiles. Here, $\sigma_{max}$ represents the privacy parameter. Modified attacks can be succinctly elucidated as described by *Gunes, Bilge & Polat (2013)*.

*Random attack*: The collection of selected items is devoid of any elements. Randomly picked placeholder elements are populated with random values. The target item is assigned a value that is randomly selected from the highest possible range.

*Average attack*: The set of objects that have been picked is also empty. Some filler items are chosen at random, and then they are filled with the item's mean value and a random value. The maximum possible random value is supplied to the target object.

*Bandwagon attack*: The popular items that are highly rated and have a high mean are the ones from which the chosen items are selected. The filler elements that are selected and picked at random are then filled with random values. The target item is given the highest possible random value, while the selected objects are given the highest possible value overall.

*Segment attack*: The model of segment attack is comparable to that of bandwagon attack. Nevertheless, products with high averages within a particular category are prioritized when selecting certain things.

*Reverse bandwagon attack*: The selection process involves choosing goods that are considered unpopular, meaning they have received low ratings and have low average scores. Filler items that have been selected and picked at random are populated with values that are also randomly selected. In this scenario, the lowest values are allocated to the selected items, while the target item is assigned the smallest random value in order to eliminate it completely.

*Love/hate attack:* There is nothing selected in the set of items. The high random values are filled in with filler items that are determined by random chance. The random value of the target object is set to its absolute minimum.

## Ant colony clustering model

In their study, *Shelokar, Jayaraman & Kulkarni (2004)* introduced a mechanism based on ant colony optimization (ACO) to cluster objects into clusters. The algorithm employs decentralized agents that emulate the foraging behavior of actual ants in order to determine the most efficient route between their nest and a food source. This algorithm's performance is evaluated by comparing it with other well-known stochastic/heuristic approaches, such as genetic algorithms, simulated annealing, and tabu search. The computational simulations demonstrate highly promising outcomes in terms of solution quality, average function evaluations, and processing time. The authors also examined the sequential stages of the ant colony optimization method used to solve clustering problems and presented the computational findings from evaluating the algorithm's performance on multiple datasets. The findings demonstrated that the ant colony optimization method surpasses other stochastic algorithms in terms of solution quality, function evaluations, and processing time.

The following is a concise overview of the ACO clustering algorithm:

- Commence the procedure by randomly allocating data points to clusters.
- Every ant transitions between data points, depositing a trail of pheromones as it goes.
- The likelihood of an ant transitioning to a specific data point is directly related to the intensity of the pheromone trail and the resemblance between the data point and the ant's current cluster.

| The symbols used in the ACO algorithm are defined below. | |
| --- | --- |
| Symbol | Meaning |
| $F$ | Fitness/Objective function |
| $N$ | Number of samples |
| $n$ | Number of attributes |
| $R$ | Number of agents |
| $S$ | Solution string |
| $K$ | Number of cluster |
| $\tau$ | Pheromone trail matrix (N x K) |
| $T$ | Number of iterations |
| $\rho$ | Persistence of trail |
| $(1 - \rho)$ | Evaporation rate |

- Clusters are updated by reassigning data points according to their current assignment and the pheromone trails.
- The pheromone trails are modified according on the quality of the existing clustering solution.

The ACO algorithm for clustering employs dispersed agents that imitate the behavior of actual ants in order to locate the most efficient route. These agents utilize pheromone trail information to provide solutions and employ local search operations to enhance the solutions. The algorithm iteratively creates new solutions, does local search operations, and updates the pheromone trail matrix for a specified number of iterations. The solution's quality is assessed based on the value of the objective function, which indicates the optimal or nearly optimal arrangement of items into clusters. The algorithm has demonstrated superior performance compared to other stochastic algorithms in terms of solution quality, function evaluations. Other studies in the literature aim to enhance the efficacy of the ant colony clustering algorithm (*Kekeç, Yumusak & Celebi, 2006*; *Liu & Fu, 2010*).

In their study, *Shelokar, Jayaraman & Kulkarni (2004)* aimed to obtain the optimal distribution of clusters in their study. This was done by reducing the sum of squared Euclidean distances between each object and its respective cluster center. This was done to attain an optimal distribution of clusters. This research utilized the clustering approach outlined in Algorithm 1. To tailor the clustering algorithm devised by *Shelokar, Jayaraman & Kulkarni (2004)* to our particular data sets, we implemented certain code modifications, which are detailed in the following paragraph. In their scientific article, *Shelokar, Jayaraman & Kulkarni (2004)* and *Kekeç, Yumusak & Celebi (2006)* presented a detailed account of the algorithm depicted in Algorithm 1.

*Shelokar, Jayaraman & Kulkarni (2004)* defined the fitness function as the total of the squared Euclidean distances between each item and the cluster's center to which it is assigned. They attempted to minimize the fitness function to achieve the optimal distribution of clusters. This study aims to determine the optimal distribution of

| | **Algorithm 1 Ant colony clustering algorithm.** |
|---|---|
| 1. | Set S, R, K, L, T $\tau\_0$ |
| 2. | $t = 1$ |
| 3. | $\tau(t) = \tau\_0$ |
| 4. | While ($t <= T$) do |
| 5. | While $i <= R$ //for all R agents |
| 6. | Construct Solution $S_i$ of all R agents using pheromone trail |
| 7. | Compute cluster centers and $F_i$ objective function values of each Si solution string |
| 8. | $i = i + 1$ |
| 9. | End |
| 10. | Sort $S_R$ solution strings and its computed $F_R$ values decreasingly |
| 11. | Select best L solutions out of R solutions using objective function values |
| 12. | While $l <= L$ //for all L local search agents |
| 13. | Construct solution strings LS again |
| 14. | Compute objective function value Fi of each LSi solution string |
| 15. | IF $F < F_l$ then replace $S_l = LS$ with LSi and $F_l = F$ |
| 16. | $l = l + 1$ |
| 17. | End |
| 18. | Update pheromone trail matrix using local search solutions |
| 19. | $\tau_{ij}(t+1) = (1 - \rho)\, \tau_{ij}(t) + \sum_{l=1}^{L} \Delta\tau_{ij}^{l}$ |
| 20. | $t = t + 1$ |
| 21. | End |
| 22. | Print best solution string ($S_1$). |

clusters by maximizing the sum of the similarities between each element and the assigned cluster center. This study employs the Pearson correlation metric, which is the prevailing similarity metric, as a substitute for Euclidean distance. The assignment procedure entails computing the similarity between each element and the cluster center, and subsequently assigning the element to the cluster based on the highest similarity value.

The dataset utilised in this study comprises an $n \times m$ user-item matrix, where n represents the number of users and m the number of items. The users are then grouped into $K$ clusters using an ant colony clustering approach. The cluster centres are calculated and a matrix of $K \times m$, symbolised by $c$, is created. The similarity of an example user a to the cluster centre ($w_{ac}$) is determined using the Pearson correlation coefficient formula, as follows:

$$w_{ac} = \frac{\sum_{j=1}^{m} \left(v_{aj} - \overline{v_a}\right)\left(v_{cj} - \overline{v_c}\right)}{\sigma_a \sigma_c} \tag{1}$$

where $c$ stands for the center of the cluster, $n \times m$ is user–item matrix, $v_{aj}$ is the rating that user $a$ gave item j, $\overline{v_a}$ and $\overline{v_c}$ are the vector mean values of user a and the corresponding cluster center respectively, and $\sigma_a$ and $\sigma_C$ are the standard deviations of user $a$ and the corresponding cluster center respectively.

## Shilling detection methods for PPCF schemes

*Shelokar, Jayaraman & Kulkarni (2004)* developed an ant colony algorithm with the goal of achieving an optimal cluster distribution by minimizing the sum of squared Euclidean distances between each object and its corresponding cluster center. This was done in order to achieve an optimal cluster distribution. In order to come up with solutions, this methodology takes into account a group of $R$ agents. An initial string representation of the solution, denoted by the letter $S$ and having a length of $N$, is created by the agent.

Previous studies have employed ant colony clustering techniques to address concerns regarding accuracy, sparsity, and scalability in non-private CF schemes (*Sobecki & Tomczak, 2010*; *Nadi et al., 2011*; *Bedi & Sharma, 2012*; *Liao, Wu & Wang, 2020*). The application of the ant colony clustering algorithm as a detection method represents a noteworthy contribution to the field, offering a novel and innovative approach to addressing these issues. This article presents a novel application of ant colony clustering for the detection of shilling attacks in privacy-preserving recommender systems. Due to the utilization of an algorithm, attack profiles are anticipated to exhibit greater similarity to one another compared to regular profiles. The attack cluster is determined to be the cluster that contains the items that are the most comparable to one another. The DegSim (Degree of Similarity with Top Neighbors) measure is employed in order to determine whether or not there is such a cluster. As a result, it is necessary to compute the DegSim value for each profile contained inside a cluster (*Chirita, Nejdl & Zamfir, 2005*). This metric is the average similarity weight with a user's top-k neighbors, and for user u the following formula can be used to determine it: DegSimu = (wu1 + wu2 +…+ wuk)/k. The cluster that has the highest average DegSim value is recognized as the attack cluster, which includes the shilling profiles. This is determined by calculating the DegSim values for all of the users that are contained within each cluster. As a consequence, this cluster is severed from the PPCF system, which results in the system being rendered incapable of being manipulated.

In order to elucidate the dynamics of the algorithm developed in the study, the triple structure (reproducibility, predictability and temporality) created by *Gore & Reynolds (2007)* was employed. Reproducibility in the context of the taxonomy refers to the repeatability of a simulation for a given set of inputs. A simulation that produces the same results every time it is run for a specific set of inputs is classified as deterministic, while a simulation that produces different results for the same inputs is considered stochastic. *Predictability* distinguishes between behaviors that can be forecasted without running the simulation and those that cannot. Predictable behaviors allow for selective sampling and enable the user to test hypotheses at critical input points, while unpredictable behaviors require running the simulation for each input to determine the observable behaviors. *Temporality* refers to the distinction between the process of achieving a final state and residing in the final state. This dimension supports different exploration methods for the

| **Algorithm 2  Ant colony clustering-based detection method on PPCF schemes.** |
|---|
| Let $U'$ represent the collection of disguised data vectors, denoted as $\{u_1, u_2, …, u_n\}$ and let C represent the set of cluster centers, denoted as $\{c_1, c_2, …, c_k\}$ |
| 1.  load Data (MLP) |
| 2.  dData ← disquse (Data) |
| 3.  Add shilling profiles to dData for all attack models |
| 4.  Set ant colony algorithm parameters: cluster size k, agent size S and iteration size t |
| 5.  For each agent, data vectors are randomly assigned to clusters. |
| 6.  Run ant colony clustering algorithm. |
| 7.  Compute the fitness values for every agent in the colony. |
| 8.  Choose the optimal fitness value, which corresponds to the cluster distribution that maximizes the total similarity between each element and the cluster center |
| 9.  Compute the DegSim value for each cluster. |
| 10. Identify the cluster with the greatest DegSim value and designate it as the attack cluster. This cluster is physically separated from the PPCF system. |
| 11. Compute precision and recall metrics to evaluate the effectiveness of the detection algorithm. |

same behavior and helps in identifying efficient methods for exploration. It distinguishes between behaviors that are constant and those that are manifested, providing guidance on the exploration process based on the behavior's temporal classification.

Upon analysis of the study according to the aforementioned triple structure, it becomes evident that disparate outcomes are yielded for identical inputs. This is due to the fact that the initial cluster distribution for all agents in the ant colony algorithm is randomly generated. Consequently, identical inputs may yield disparate outcomes. The scheme developed in this study is stochastic in terms of reproducibility. Since random number distributions are employed to conceal the data sets, different unexpected results may be observed in each simulation. To mitigate the impact of these unforeseen outcomes, the simulations are conducted multiple times and the results are averaged. When the scheme in the study is analysed in terms of predictability, it is unpredictable due to the potential for unexpected results. In terms of temporality, the scheme can be defined as manifested, as it supports diverse exploration methods for the same behaviour.

## Experiments

Several sets of experiments are conducted on actual data sets to demonstrate the efficacy of the shilling attack detection technique in PPCF schemes, specifically in relation to six-shilling attack models, when applied to disguised databases. The effectiveness of shilling attacks relies on two key factors: the size of the filler and the size of the attack. Filler size refers to the proportion of empty cells that are occupied in the attacker's profile. The attack size parameter determines the quantity of attack profiles to be injected. Thus, the severity of an attack is directly correlated with the quantity of users in the system. For example, if the attack size is ten percent, it means there would be 100 attack profiles targeting a system that initially hosts 1,000 users. Privacy-preserving control settings are maintained constant, $\beta_{max} = 25\%$ and $\sigma_{max} = 2$.

## Data set and evaluation criteria

The MovieLens public data set (MLP) and Jester are utilized in the experiments. The MLP data set was gathered by the GroupLens research team, as published on their website (http://www.grouplens.org). The dataset consists of 100,000 ratings given by 943 individuals for 1,682 movies. The ratings inside the set are distinct, ranging from 1 to 5. Every user evaluates a minimum of 20 films. The Jester data collection was made available by the Jester Joke Recommender System through their website (http://eigentaste.berkeley.edu/dataset/). Jester incorporates a scale of numerical continuous ratings that spans from −10 to 10. The density is approximately 56%. The set consists of 73,496 users and 100 jokes. While we utilized the entirety of users' data in MLP, we employed a random sampling technique to select a subset of 1,000 users from the Jester dataset.

The usual metrics of precision and recall are employed to evaluate the performance of detection systems. The fundamental definition of such measures is provided as follows:

The precision of a model is determined by the ratio of true positives to the sum of true positives and false positives.

$$Precision = Number\ of\ true\ positives/(Number\ of\ true\ positives + Number\ of\ false\ positives)$$

Recall is determined by dividing the count of true positives by the sum of true positives and false negatives.

$$Recall = Number\ of\ true\ positives/(Number\ of\ true\ positives + Number\ of\ false\ negatives)$$

Given our main focus on the efficacy of the algorithms in detecting potential attacks, we closely monitor each of these indicators in relation to attack identification. The number of true positives refers to the count of accurately identified attack profiles, whereas the number of false positives represents the quantity of genuine profiles that are mistakenly categorized as attack profiles. On the other hand, the number of false negatives corresponds to the number of attack profiles that are inaccurately classed as genuine profiles.

## METHODOLOGY

In the following way, the fundamental concepts that underlie our methodology for conducting experiments can be explained in further detail. At the outset, two distinct collections of target objects are compiled. In MLP data, each of them offers defense against push and nuke strikes for a total of 50 movies. Due to the limited number of jokes in Jester, target item sets for push and nuke attacks include 25 jokes. The selection of objects is done in a random fashion using stratified sampling. Instinctually, it is deemed unreasonable to make an effort to promote something that is generally loved or to get rid of something that is generally hated. During the trials, every test user with access to the system will launch attacks against all of the predefined targets. For each assault model, the precision and recall figures are produced to indicate how good the detection approaches are. The reason we did not carry out a segment attack in Jester is due to the absence of a joke category in Jester.

## Empirical results

### Effect of filler size parameter

Effects experiments are carried out in order to demonstrate the effectiveness of the detection methods while detecting fraudulent profiles within masked databases using a range of different filler size values. Because filler items provide the foundation for leaking into the neighborhoods of legitimate users throughout the recommendation process, the size of the filler item pool has a direct correlation with the success of an assault that has been carried out. While the attack size, $\beta_{max}$, and $\sigma_{max}$ remains the constant at 15%, 25%, 2, respectively, the filler size might be anything from 5% to 50% of the total.

The ant colony clustering based detection approach utilizes profile similarities to conduct clustering. Given that all attack models are constructed using specific methods, it is inherent that they exhibit similarities to one another. The ant colony clustering-based detection approach identifies the most closely related cluster and separates it from the system. As the magnitude of the filler size value grows, the attack profiles exhibit a greater resemblance to the actual profiles. Consequently, the cluster of attack profiles will contain a greater number of genuine profiles. In this scenario for both data sets, a greater number of authentic profiles are removed from the system, resulting in a fall in the precision values. Augmenting the filler size values solely amplifies the quantity of authentic profiles within the search cluster while diminishing the precision value. Nevertheless, with an increase in filler size, the counterfeit profiles exhibit a greater resemblance to the genuine profiles, resulting in a more pronounced overlap between the two clusters. In this instance, the recall value exhibited a decline as the filler size increased, similar to the pattern observed in precision.

Precision and recall values for Bandwagon, Segment and Reverse Bandwagon attacks are slightly higher than other attack models. The reason for this is that the same values are given to certain item groups (popular, unpopular items, *etc.*) in these attack models, which increases the similarity of these profiles. Ant colony clustering algorithm can group these attack patterns better than others. In Table 1, the MLP dataset demonstrates a significant decrease in precision and recall values for all attack models when the filler size is set at 50%. Regarding the Jester dataset in Table 2, when the filler size is 50%, the precision and recall values experience a lesser decrease, with the exception of the Random attack. For the Jester dataset, the Bandwagon and Reverse Bandwagon attack models demonstrated more success compared to other models, similar to MLP. The precision and recall values exhibited a negative correlation with the filler size values across all attack models.

When the precision values are less than 1.0, some authentic profiles are mistakenly classified as attack profiles. Similarly, when the recall values are less than 1.0, some attack profiles are incorrectly labeled as genuine profiles.

### Effect of attack size parameter

Multiple sets of experiments were carried out to examine the effectiveness of shilling attack detection methods on private environments, while varying the size values of the attacks. The significance of attack magnitude underscores the need of identifying the quantity of fraudulent profiles to be introduced into a database, as well as the practical implications of

**Table 1 Performance of ant colony clustering with varying filler size values (MLP).**

|  | Precision | | | | | Recall | | | | |
|---|---|---|---|---|---|---|---|---|---|---|
| Filler size | 5 | 10 | 15 | 25 | 50 | 5 | 10 | 15 | 25 | 50 |
| Random | 0.906 | **0.918** | 0.817 | 0.108 | 0.009 | **0.994** | 0.981 | 0.803 | 0.101 | 0.012 |
| Average | 0.867 | **0.900** | 0.523 | 0.057 | 0.002 | **0.938** | 0.916 | 0.526 | 0.086 | 0.002 |
| Bandwagon | 0.901 | 0.907 | **0.933** | 0.747 | 0.003 | **0.999** | 0.989 | 0.948 | 0.690 | 0.005 |
| Segment | 0.934 | 0.926 | **0.937** | 0.653 | 0.002 | **0.999** | 0.972 | 0.981 | 0.617 | 0.003 |
| RB | 0.928 | **0.936** | 0.929 | 0.926 | 0.001 | **0.999** | 0.999 | 0.975 | 0.953 | 0.001 |
| Love/Hate | **0.904** | 0.877 | 0.531 | 0.028 | 0.027 | **0.927** | 0.810 | 0.442 | 0.033 | 0.035 |

Note:
The best outcomes are given in bold.

**Table 2 Performance of ant colony clustering with varying filler size values (Jester).**

|  | Precision | | | | | Recall | | | | |
|---|---|---|---|---|---|---|---|---|---|---|
| Filler size | 5 | 10 | 15 | 25 | 50 | 5 | 10 | 15 | 25 | 50 |
| Random | 0.726 | **0.778** | 0.626 | 0.446 | 0.052 | 0.805 | **0.836** | 0.615 | 0.423 | 0.051 |
| Average | 0.755 | 0.792 | **0.811** | 0.786 | 0.672 | **0.877** | 0.861 | 0.865 | 0.830 | 0.687 |
| Bandwagon | 0.797 | **0.799** | 0.775 | 0.737 | 0.720 | 0.983 | 0.983 | **0.995** | 0.991 | 0.901 |
| Segment | – | – | – | – | – | – | – | – | – | – |
| RB | 0.802 | **0.882** | 0.820 | 0.801 | 0.710 | 0.984 | 0.985 | 0.950 | **0.993** | 0.910 |
| Love/Hate | 0.751 | **0.824** | 0.765 | 0.768 | 0.704 | 0.922 | **0.927** | 0.892 | 0.867 | 0.819 |

Note:
The best outcomes are given in bold.

such assault. The attack size determines a balance between the detectability and the impact of the attack model being used. Hence, we conducted trials by systematically altering the attack size within the range of 1% to 15%, while maintaining a fixed filler size of 25%. Tables 3 and 4 provide the average precision and recall values for the ant colony clustering-based detection strategy, with varied attack size values, for MLP and Jester, respectively.

According to the data presented in Tables 3 and 4, when the attack size increases for both datasets, the concentration of attack profiles in the cluster that needs to be separated from the system also increases. In this scenario, the system will isolate a greater number of attack profiles and a smaller number of genuine profiles, as the majority of the cluster will be comprised of attack profiles. Therefore, experiments conducted with larger attack size values yield improved precision and recall values. Once the most compact group is identified and separated from the database, the genuine profiles can be eliminated from the system. The precision value decreases when real profiles are removed from the user-item matrix. As a result, at low attack size, fake profiles are more likely to be scattered in different clusters. After the attack size exceeds 10%, a cluster consisting mostly of fake profiles can be formed. Therefore, at low attack size, both the precision and recall values are lower due to the distribution of fake profiles into different clusters. Both data sets yield satisfactory precision and recall values when the assault size is 10% or greater. This is because a specific number of attack profiles is necessary to form an attack cluster. When

**Table 3 Performance of ant colony clustering with varying attack size values (MLP).**

| | Precision | | | | | Recall | | | | |
|---|---|---|---|---|---|---|---|---|---|---|
| Attack Size | 1 | 3 | 5 | 10 | 15 | 1 | 3 | 5 | 10 | 15 |
| Random | 0.005 | 0.010 | 0.015 | 0.008 | **0.099** | 0.120 | 0.053 | 0.052 | 0.016 | **0.092** |
| Average | 0.001 | 0.002 | 0.001 | 0.001 | **0.540** | 0.010 | 0.013 | 0.002 | 0.002 | **0.080** |
| Bandwagon | 0.003 | 0.004 | 0.002 | 0.181 | **0.905** | 0.050 | 0.030 | 0.008 | 0.188 | **0.809** |
| Segment | 0.001 | 0.007 | 0.004 | 0.182 | **0.650** | 0.010 | 0.043 | 0.016 | 0.198 | **0.610** |
| RB | 0.001 | 0.003 | 0.000 | 0.446 | **0.929** | 0.020 | 0.017 | 0.000 | 0.476 | **0.944** |
| Love/Hate | 0.004 | 0.012 | 0.018 | 0.016 | **0.520** | **0.090** | 0.073 | 0.066 | 0.033 | 0.027 |

Note:
The best outcomes are given in bold.

**Table 4 Performance of ant colony clustering with varying attack size values (Jester).**

| | Precision | | | | | Recall | | | | |
|---|---|---|---|---|---|---|---|---|---|---|
| Attack Size | 1 | 3 | 5 | 10 | 15 | 1 | 3 | 5 | 10 | 15 |
| Random | 0.007 | 0.016 | 0.052 | 0.371 | **0.588** | 0.100 | 0.070 | 0.122 | 0.438 | **0.563** |
| Average | 0.004 | 0.006 | 0.138 | 0.645 | **0.779** | 0.060 | 0.027 | 0.322 | 0.849 | **0.849** |
| Bandwagon | 0.001 | 0.135 | 0.359 | 0.631 | **0.769** | 0.010 | 0.553 | 0.798 | 0.959 | **0.972** |
| Segment | – | – | – | – | – | – | – | – | – | – |
| RB | 0.007 | 0.087 | 0.449 | 0.720 | **0.823** | 0.100 | 0.243 | 0.880 | **0.980** | 0.979 |
| Love/Hate | 0.011 | 0.041 | 0.109 | 0.534 | **0.732** | 0.130 | 0.193 | 0.292 | 0.652 | **0.850** |

Note:
The best outcomes are given in bold.

the attack sizes are small, fake profiles can be dispersed among diverse clusters. The precision and recall values for bandwagon, segment, and reverse Bandwagon attacks demonstrate a clear superiority over other attack models, as stated in the preceding section. This occurs because these attack models assign identical values to specific categories of items (such as popular or unpopular items), thereby heightening the resemblance of these profiles.

## COMPARISON AND DISCUSSION

The findings of this study are juxtaposed with three distinct clustering techniques that have been previously established in the academic literature. The algorithms used are bisection k-means, hierarchical clustering, and ordinary k-means.

*Bilge, Ozdemir & Polat (2014)* introduced a new way for detecting shilling attacks that are highly specific. The method is based on a *bisecting k-means* clustering methodology, where attack profiles are collected in a leaf node of a binary decision tree. *Gunes & Polat (2015)* proposed a hierarchical clustering-based method for detecting shilling attacks in privacy-preserving recommendation schemes. They aimed to detect shilling profiles in privacy-preserving collaborative filtering systems, providing a dependable method for isolating these profiles. *Gunes & Polat (2016)* used four different attack detection methods including k-means algorithm to filter out fake profiles generated by six well-known shilling

**Table 5 Comparison of the clustering algorithms in private cases (MLP).**

| | Precision | | | | Recall | | | |
|---|---|---|---|---|---|---|---|---|
| Algorithm | Bi-section | Hierarcial | k-means | Ant colony | Bi-section | Hierarcial | k-means | Ant colony |
| Random | – | 0.003 | 0.245 | 0.108 | – | 0.002 | 1.000 | 0.101 |
| Average | 0.982 | 0.918 | 0.308 | 0.057 | 0.916 | 0.830 | 1.000 | 0.086 |
| Bandwagon | 0.947 | 1.000 | 0.260 | 0.747 | 0.987 | 0.999 | 1.000 | 0.690 |
| Segment | 1.000 | 1.000 | 0.297 | 0.653 | 1.000 | 0.967 | 1.000 | 0.617 |
| RB | – | 1.000 | 0.290 | 0.926 | – | 1.000 | 1.000 | 0.953 |
| Love/Hate | – | 0.640 | 0.231 | 0.028 | – | 0.329 | 1.000 | 0.033 |

attacks on perturbed data. They evaluated these detection methods with respect to their ability to identify bogus profiles.

A comparison of the precision metric reveals that the hierarchical and bisection k-means algorithms yield the most successful outcomes. The ant colony algorithm outperforms the k-means algorithm. A high sensitivity value indicates that attack profiles and real profiles have been successfully grouped. A sensitivity metric of 1.0 indicates that the cluster isolated from the system consists only of attack profiles, while a sensitivity metric of less than 1.0 indicates that attack profiles and real profiles are grouped in the same cluster and isolated from the system. When the results in Table 5 are analysed, hierarchical clustering and bisection clustering achieved the highest precision metric value by separating real profiles and fake profiles and grouping them in different clusters. In all methods, lower precision metric values were obtained in random and love/attack attack models. This is due to the fact that random numbers are used to create profiles in these attack models, which results in profiles being less similar to each other and therefore distributed in different clusters. The ant colony algorithm performed successful groupings, in particular against bandwagon, segment and reverse bandwagon attack models. The k-means method was unsuccessful in terms of the precision metric due to the clustering of a significant number of genuine profiles with those of the attack profiles. This resulted in the elimination of the genuine profiles from the system.

The k-means method, which was unsuccessful in terms of the precision metric, obtained the most successful results in terms of the Recall metric. This result demonstrates that the k-means method assigns all fake profiles to a single cluster and successfully isolates them from the system. Recall values below 1.0 indicate that some fake profiles are scattered in clusters with real profiles and remain in the system. Although the ant colony algorithm achieved high recall values, particularly for bandwagon, segment and reverse bandwagon attacks, it was less successful than other methods. The results demonstrate that the ant colony clustering-based detection method is unable to collect all attack profiles within the isolated cluster, with some remaining in different clusters. The low precision is attributed to the fact that both real and fake profiles are isolated from the system. The ant colony algorithm demonstrated particular efficacy in the identification of bandwagon, segment, and reverse bandwagon attack types.

## CONCLUSION

The optimal number of clusters in the ant colony method can be discovered through empirical tests using the relevant data. Choosing the appropriate cluster and separating it from the system is of utmost significance. Excluding an incorrect cluster from the system may result in the exclusion of numerous genuine profiles. Nevertheless, numerous attack profiles will persist within the system. Given that the attack generating algorithms generate comparable profiles, it follows that all attack models will yield similar outcomes. The ant colony clustering algorithm utilizes this similarity to execute clustering. In order to identify the cluster that needs to be separated, we employed the DegSim measure, which calculates the average similarity weight between a user and its k most similar neighbors. The neighbors in the clustering technique are chosen as constituents of the respective cluster. The cluster with the highest DegSim metric value is chosen as the assault cluster and separated from the system.

Overall, the detection method's performance declines as the filler size values increase for both datasets. The correlation between genuine profiles and attack profiles grows as the filler size increases, leading to this outcome. Our approach has a higher probability of grouping genuine profiles and fraudulent profiles together when the filler size is increased. Consequently, the values of both precision and recall metrics decline as the size of the filler increases.

Upon examining the overall results, it is evident that as the magnitude of the attack increases, both precision and recall values also increase. This is because, at smaller attack sizes, the clustering algorithm has a lower probability of forming a cluster with a significant concentration of attack profiles. Once a specific attack size is attained, the system has the capability to group the attack profiles together within a cluster. Our experimental findings indicate that as the attack size surpasses 10 percent, there is a noticeable increase in both precision and recall values.

The ant colony clustering algorithm is characterized by its high processing cost and potential suboptimal performance when used to datasets with a large number of dimensions. Future research endeavors aim to develop methodologies that can enhance performance speed. Furthermore, we intend to employ different clustering algorithms as detection mechanisms within a privacy-preserving recommender system.

### Funding

The author received no funding for this work.

### Competing Interests

The authors declare that they have no competing interests.

## Author Contributions

- Ihsan Gunes conceived and designed the experiments, performed the experiments, analyzed the data, performed the computation work, prepared figures and/or tables, authored or reviewed drafts of the article, and approved the final draft.

## Data Availability

The code and data are available in the Supplemental Files.

The JESTER dataset is available at: https://eigentaste.berkeley.edu/dataset.

The MLP dataset is available at: https://grouplens.org/datasets/movielens/100k.

## Supplemental Information

Supplemental information for this article can be found online at http://dx.doi.org/10.7717/peerj-cs.2137#supplemental-information.

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
