# Peer review of "A novel clustered-based detection method for shilling attack in private environments"

_PeerJ Computer Science, doi:10.7717/peerj-cs.2137_

## Round 0.1 · original submission · Major Revisions

I have received reviews of your manuscript from scholars who are experts on the cited topic. They find the topic very interesting; however, several concerns must be addressed regarding experimental results and comparisons with current approaches. These issues require a major revision. Please refer to the reviewers’ comments listed at the end of this letter, and you will see that they are advising that you revise your manuscript. If you are prepared to undertake the work required, I would be pleased to reconsider my decision. Please submit a list of changes or a rebuttal against each point that is being raised when you submit your revised manuscript.

Thank you for considering PeerJ Computer Science for the publication of your research.

With kind regards,

Prof. Vicente Alarcon-Aquino, PhD (Lond), DIC
Academic & Section Editor
PeerJ Computer Science

Reviewer 1 ·

Basic reporting

This work targets on an important privacy problem. It addresses the increasingly critical issue of privacy-preserving collaborative filtering (PPCF) within the realm of recommender systems, specifically focusing on its vulnerability to shilling or profile injection attacks. These attacks involve malicious users injecting fake profiles into the system to manipulate recommendations, either to unduly promote or demote certain products. Despite the significance of PPCF in protecting user preferences and data in online shopping and electronic commerce, there has been a notable gap in research concerning the identification and mitigation of shilling attacks within these privacy-oriented systems. The authors point out that previous studies mainly focuses on collaborative filtering algorithms that do not prioritize privacy. Hence, they propose applying an ant colony clustering detection method on this specific problem. However, there are some important issues in this paper, I will discuss them in details.

Experimental design

Motivation issues:The authors claim there are a lot of shilling attack detection work focuses on CF systems, but not on PPCF systems. That's a good point, but the authors don't explain very well on the differences of detecting shilling attacks between CF and PPCF systems. Why are PPCF more challenging? Why existing detection techniques cannot be applied to PPCF systems? What are the main challenges of detecting shilling attacks in PPCF systems? These questions should be the main motivation of this work, and should be discuss thoroughly in both introduction and motivation section(if has one). Besides, the related work section spends too many spaces discussing previous works in CF systems, only one paragraph discussing works PPCF systems. I would recommend the authors adding more discussion on works in PPCF systems and clarify the limitations of these works. That should be another key motivation for this paper.

Design issues: It seems to this paper simply uses ant colony clustering on this particular problem without explaining why this algorithm is particularly suitable for this problem. There are many other clustering algorithms and why don't choose those? Also, is there any novel improvement on traditional ant colony clustering algorithm? More details and clarifications are needed in this paper to make this work convincing.

Validity of the findings

See the comments above.

Additional comments

The writing needs to be improved:
For example, in line 146: "Few studies in the literature focus on identifying attacks against privacy-preserving recommender systems. In their study, Gunes & Polat, (2016) specifically examined the identification of shilling attacks in privacy-preserving collaborative filtering systems." It is strange to start with "in their study" without mentioning the work in previous sentences.
Also, in line 293, "has been to address issues.."

The format of this paper is terrible:
The tables and figures also need to be improved. Table 2, 3, 4 lacks the last line. Figure 1 is really unclear. Also please rewrite the Algorithm 1 in a formal algorithm format.

·

Basic reporting

The paper uses clear and unambiguous, and professional English used throughout. The background/context is provided and well established. The literature references describing related work however, could be improved. There are at least two recent approaches that are related to the work described by the authors. These papers are identified below:

Ben Horin, A., & Tassa, T. (2021, September). Privacy preserving collaborative filtering by distributed mediation. In Proceedings of the 15th ACM Conference on Recommender Systems (pp. 332-341).

Kashani, S. M. Z., & Hamidzadeh, J. (2020). Feature selection by using privacy-preserving of recommendation systems based on collaborative filtering and mutual trust in social networks. Soft Computing, 24(15), 11425-11440.

The article structure is professional. However, the presentation of the tables and figures can be improved. Decimals instead of commas should be used for the numeric data in tables 1-4. In table 5 the style used to represent numeric data are mixed; both commas and decimals are used. This should be changed to use decimals throughout. In addition, Figure 1 is extremely blurry and difficult to hread. In addition, the style of the entries in the reference section is inconsistent. For example the entry of the paper by 'KEKEÇ, G'. has the authors name in all capital letters while no other entries do.

The results are self-contained and relevant to the contributions identified by the author. However, the paper would be improved if these contributions were rephrased as research questions with specified hypotheses. There are terms used in the equation on line 280 that are undefined. It is not specified what wac, a, c, and m are in the equation.

Experimental design

The original primary research presented in the paper is within Aims and Scope of the journal. However, the paper does not explicitly identify research questions. Instead three contributions are itemized. The paper would be improved if these contributions were rephrased as research questions with specified hypotheses. The contributions do establish how the research fills an identified knowledge gap.

The investigation is performed to a high ethical standard. However, the technical standard to which it is employed could be improved. Specifically, this study compares the detection methods employed with other approaches previously utilized in the literature. However, this comparison (in Table 5) only identifies the precision and recall of 4 approaches. It does not establish if there are any statistically significant differences in the level of performance of these techniques. The paper would be improved by performing tests for statistical significance among the compared techniques to highlight which of the approaches, with respect to precision and recall, are materially different from one another. In addition, it is unclear why the Book-Crossing dataset was not included in the evaluation. This is an established data set for the Privacy-Preserving Collaborative Filtering System problem as highlighted below.

Açıl, E. T., & Yargıç, A. (2022). Privacy-Preserving Collaborative Filtering System For Book-Crossing Dataset. In VI. International European Conference on Interdisciplinary Scientific Research. IKSAD.

The methods in the paper are not described in sufficient detail. There is insufficient information, including what is provided in the supplementary material to replicate the work. First, identifying the characteristics of the Ant colony clustering simulation algorithm in an established taxonomy would improve the clarity of the proposed approach. Specifically highlighting these properties in the 3-tuple structure for simulation algorithms established by Gore and Reynolds in 2007 would make explicit how the dynamics of the algorithm work. It appears it is (stochastic, unpredictable, manifested). Highlighting this will make it clear how the algorithm works to readers.


Gore, R., & Reynolds, P. F. (2007, December). An exploration-based taxonomy for emergent behavior analysis in simulations. In 2007 Winter Simulation Conference (pp. 1232-1240). IEEE.

In addition, the paper would be improved by including explicit instructions (i.e README files) in the supplementary materials that describe how to replicate each of the pieces of the evaluation (i.e. results in Tables 1-5). As things are currently packaged it is not clear how the reader/reviewer would run the code, what data is needed in which places, what are the system requirements, what version of matlab is used, what is needed for each piece of the evaluation.

Validity of the findings

As previously described currently it is not possible to replicate the results of the study. the paper would be improved by including explicit instructions (i.e README files) in the supplementary materials that describe how to replicate each of the pieces of the evaluation (i.e. results in Tables 1-5).

In addition, it is not clear that the underlying data have been provided. I could not find the data files produced for the tables in 1-5 in the supplementary information. Perhaps this is because it was unclear how I would produce them, but they also should be provided to highlight the target that the code, when run, should match. Doing so would make the replication of the results and data more robust. In addition, it is not clear that the comparison of the approach against alternative techniques presented in table 5 is statistically sound. As previously, identified the paper would be improved by performing tests for statistical significance among the compared techniques to highlight which of the approaches, with respect to precision and recall, are materially different from one another.

The conclusions of the paper are appropriately stated, connected to the original question investigated, and limited to those supported by the results.

---

## Round 0.2 · accepted · Accept

I am pleased to inform you that your work has now been accepted for publication in PeerJ Computer Science.

Please be advised that you are not permitted to add or remove authors or references post-acceptance, regardless of the reviewers' request(s).

Thank you for submitting your work to this journal. On behalf of the Editors of PeerJ Computer Science, we look forward to your continued contributions to the Journal.

With kind regards,

·

Basic reporting

My concerns related to basic reporting have been sufficiently addressed. Clear and unambiguous, professional English is used throughout the manuscript. Literature references and sufficient field background/context are provided. The manuscript has professional article structure, figures, tables. The raw data is shared. The reporting is self-contained with relevant results to hypotheses. All formal results include clear definitions of all terms and theorems, and detailed proofs.

Experimental design

My concerns related to the experimental design have been sufficiently addressed. The paper contains original primary research within Aims and Scope of PeerJ Computer Science. The research questions are well defined, relevant and meaningful. It is stated how research fills an identified knowledge gap. The investigation in the manuscript is rigorous and performed to a high technical & ethical standard. The methods described with sufficient detail & information to replicate.

Validity of the findings

My concerns related to the validity of the findings have been sufficiently addressed. There is meaningful replication and the rationale / benefit to literature is clearly stated. All underlying data have been provided; they are robust, statistically sound, & controlled. The conclusions are well stated, linked to original research question and limited to the supporting results.

Additional comments

All my concerns have been sufficiently addressed. The paper is now suitable for publication.